# Lagged Association of Ambient Outdoor Air Pollutants with Asthma-Related Emergency Department Visits within the Pittsburgh Region

**DOI:** 10.3390/ijerph17228619

**Published:** 2020-11-20

**Authors:** Brandy M. Byrwa-Hill, Arvind Venkat, Albert A. Presto, Judith R. Rager, Deborah Gentile, Evelyn Talbott

**Affiliations:** 1Department of Environmental and Occupational Health, University of Pittsburgh Graduate School of Public Health, Pittsburgh, PA 15261, USA; 2Department of Emergency Medicine, Allegheny Health Network, Pittsburgh, PA 15212, USA; Arvind.Venkat@ahn.org; 3Center for Atmospheric Particle Studies and Department of Mechanical Engineering, Carnegie Mellon University, Pittsburgh, PA 15213, USA; apresto@andrew.cmu.edu; 4Department of Epidemiology, University of Pittsburgh Graduate School of Public Health, Pittsburgh, PA 15213, USA; JNRST8@pitt.edu (J.R.R.); EOT1@pitt.edu (E.T.); 5Allergy and Asthma Wellness Centers, Butler, PA 16066, USA; dgentile@cpasthma.org

**Keywords:** asthma, pollution, emergency department visits, pediatrics, adults, ozone

## Abstract

Asthma affects millions of people globally and is especially concerning in populations living with poor air quality. This study examines the association of ambient outdoor air pollutants on asthma-related emergency department (ED) visits in children and adults throughout the Pittsburgh region. A time-stratified case-crossover design is used to analyze the lagged effects of fine particulate matter (PM_2.5_) and gaseous pollutants, e.g., ozone (O_3_), sulfur dioxide (SO_2_), nitrogen dioxide (NO_2_), and carbon monoxide (CO) on asthma-related ED visits (*n* = 6682). Single-, double-, and multi-pollutant models are adjusted for temperature and analyzed using conditional logistic regression. In children, all models show an association between O_3_ and increased ED visits at lag day 1 (OR: 1.12, 95% CI, 1.03–1.22, *p* < 0.05) for the double-pollutant model (OR: 1.10, 95% CI: 1.01-1.20, *p* < 0.01). In adults, the single-pollutant model shows associations between CO and increased ED visits at lag day 5 (OR: 1.13, 95% CI, 1.00–1.28, *p* < 0.05) and average lag days 0–5 (OR: 1.22, 95% CI: 1.00–1.49, *p* < 0.05), and for NO_2_ at lag day 5 (OR: 1.04, 95% CI: 1.00–1.07, *p* < 0.05). These results show an association between air pollution and asthma morbidity in the Pittsburgh region and underscore the need for mitigation efforts to improve public health outcomes.

## 1. Introduction

Asthma is a common chronic lung disease with varying phenotypes, some of which may be worsened by environmental factors such as air pollution. Air pollution is a complex mixture of gases and particulate matter (PM) that varies in concentration across the United States. The variability in concentration and composition is due to differing weather patterns and pollution sources [1]. Air pollution also varies from day to day and season to season within a region [2]. Numerous studies have associated air pollutants with adverse health outcomes, such as asthma exacerbations [3,4,5,6,7,8,9]. Short-term exposure to PM_2.5_ (PM smaller than 2.5 μm in diameter), O_3_, and other gaseous pollutants such as SO_2_, NO_2_, and CO (common products associated with the burning of fossil fuels and industrial emissions) have been shown to trigger asthma exacerbations and result in increased emergency department (ED) admissions in both children and adults [10,11] O_3_ is unique from the other pollutants since it is not a primary pollutant. Rather, it is a secondary pollutant formed by photochemical reactions between sunlight and pollutant precursors, such as volatile organic compounds and nitrogen oxides. In some instances, the literature on health effects from air pollution is inconsistent and thus warrants further investigation [12]. The U.S. Environmental Protection Agency (EPA) sets the National Ambient Air Quality Standards for criteria pollutants based in part on controlled human exposure studies assessing airway hyper-responsiveness (AHR) [13]. These controlled exposure studies help understand how the lung responds to air pollutants.

Pittsburgh, located in southwestern Pennsylvania, has a unique topography because its metropolis is built within the hilly Appalachian Mountains [14]. According to the American Lung Association’s 2020 annual air quality report, Pittsburgh ranks poorly in the country for year-round particle pollution and O_3_ [15], thus making this region a good location to study pollution-related health effects. A previously-published study in Pittsburgh examined the potential influence of gender and ethnicity on asthma-related ED visits between 2002 and 2005 and found O_3_ and PM_2.5_ to be significantly associated with an increased likelihood of ED visits for both Black and White Americans [14]. This prior study used a two-pollutant model and did not assess differential effects based on age. Other studies have found similar associations between ED visits and O_3_ [16,17,18]; however, few of these studies included five pollutant models. The primary objective of this study was to examine the association between ambient air pollution and the risk of asthma-related ED visits in both children and adults using a five-pollutant model.

## 2. Materials and Methods

A time-stratified case-crossover study design with conditional logistic regression was used to examine the short-term associations of fine particulate matter (PM_2.5_) and gaseous pollutants with asthma ED visits. The case-crossover design is commonly chosen to investigate transient effects on the risk of acute health events [19]. Specifically, this method is frequently used in epidemiology studies to examine the short-term effects of air pollution on respiratory disease events [20]. This design uses defined cases only and compares an individual’s exposure experience just prior to the event with exposure at other times (the referent periods). An advantage of the case-crossover design is that, since each case serves as its own control, confounding by risk factors that are invariant (such as age, sex, and race) or slowly-changing (such as tobacco smoke exposure and socio-economic status) are controlled by the design.

Data on ED visits from Allegheny Health Network (AHN), a six-hospital regional health system, with a primary discharge diagnosis of asthma from 1 July 2008 to 30 June 2013 were included in our analysis. All have hospital-based emergency departments staffed by emergency physicians with up-to-date diagnostic and treatment capabilities. The visits to each of these EDs ranged from 25,000 to 55,000 per year. Data for each visit included: name of hospital; pseudo ID; dates of admission and discharge; primary discharge diagnosis and up to three secondary diagnoses; zip code of residence; age, gender, and race; and disposition from ED (whether or not an individual was subsequently admitted as an in-patient or discharged to home). Specifically included in these analyses were visits with a primary discharge diagnosis of asthma, defined as ICD-9 Codes 493.XX. Analyses were limited to Allegheny County residents, ages five years and older.

### 2.1. Environmental Exposure Measures

For this study, ambient pollutant data were downloaded from the Environmental Protection Agency (EPA) website and included concentrations and air quality index (AQI) values. A reference monitor in urban Pittsburgh (air quality system (AQS) ID: 420030038) measured PM_2.5_, O_3_, NO_2_, and CO. SO_2_ measurements were obtained from a separate reference monitor in Avalon, PA (AQS ID:420030002). This monitor was used as an indicator of regional SO_2_ air pollution, since SO_2_ was not measured at the Pittsburgh location. These monitoring sites are part of the EPA AQS that is used to monitor compliance with the Clean Air Act.

These reference monitors were selected because they measured the pollutants of interest, provided temporal data, and were centrally located between the six hospitals where asthma ED visit data were collected (Figure 1). To address missing values from the monitors due to technical errors in their functioning, we imputed missing values using the mean of the nearest valid values (one before and one after) to fill gaps in the data. The validation of this imputation has been deemed an acceptable process and has been scrutinized in many studies [15,16,21,22,23,24,25].

Meteorological variables, including the daily minimum and maximum air temperatures, were obtained from the Pittsburgh International Airport for dates between July 2008 and June 2013. These data were downloaded from the National Oceanic Atmospheric Association (NOAA) Climate Data online website. Average daily temperature was defined as the average of the minimum and maximum air temperatures for that day. Similar to the pollutant monitoring, missing values were imputed from the nearest days’ values.

### 2.2. Statistical Analysis

For our analyses, the level of ambient air pollution at the time of day zero (day of admission) as well as lagged days just before an individual visited the ED were compared with levels at referent times. This approach is commonly used to evaluate the acute effects of transient air pollution exposures by comparing outcome risks in the same individual at different times. We used a time-stratified approach to select referent periods in 7-day intervals to minimize confounding factors such as day of the week, which could be associated with adverse pollution events. With this approach, the overall time period is divided into strata and exposure in the time period just prior to the event, and exposures in multiple reference periods are compared within that stratum of time. For this study, we used a 28-day strata and referent periods of 7, 14, and 21 days (either before or after the ED event) within each stratum. To assure independence of events within strata, a washout period of 7 days was used to remove recurrent events (ED visits) for an individual.

We evaluated the effect of exposure to pollutants on the day of the ED visit and prior days; lag 0 represents exposure on the same day as the ED visit. We also examined the effects of cumulative days of exposure to calculate average exposures over several days. The average of lag days 0–5 was calculated as the mean of lag days 0, 1, 2, 3, 4, and 5, respectively. Pollutant effects were examined with models containing just one single day or an average of lag days 0–5. We controlled for temperature effects in all models by including average temperature at the same lag(s) as the pollutant(s). Because case-crossover is so tightly controlled by the referent definition of “within 28 days”, cubic splines was not used. In addition to the single-pollutant models for PM**_2.5_**, O**_3_**, NO**_2_**, SO**_2_**, and CO, we conducted two-pollutant models for PM**_2.5_** and O**_3_** adjusted for temperature. Lastly, we conducted multi-pollutant models with all five pollutants. The results are presented as odds ratios (ORs) and the associated 95% confidence intervals for every 10 μg/m^3^ increase in PM**_2.5_**; 10 ppb for O**_3_**, NO**_2_**, and SO**_2_**; and 1 ppm for CO. Only significant findings are included within the text; however, the full data set may be found within Appendix A.

Data analyses were conducted using the case-crossover tool (C-CAT) developed by Apex Epidemiology Research in collaboration with the New York State Department of Health for use with SAS (Abraham JH and Bateson TF, 2016). C-CAT is public domain software that provides an easy-to-use interface for SAS code to implement the time-stratified case-crossover analysis. Separate analyses were conducted for children (ages 5–17 years) and for adults (ages 18 years and older).

## 3. Results

The study population distribution by age and sex is shown in Figure 2. The majority of ED cases within our sample population were adults (*n* = 6682). Specifically, 87% were adults (*n* = 5842) and 13% were children (*n* = 840). Overall, there were higher rates of ED visits among women compared to men.

### 3.1. Air Pollution Data

The time scale for each pollutant recorded varies and is dependent upon EPA reporting requirements. The odds of an asthma ED visit related to an increase in exposure per increase of 10 (unit dependent on pollutant as referenced in the methods section) for a single day was significant at different lags for different pollutants. The strongest associations between pollutant levels and ED visits were observed for O_3_ within single-, double-, and five-pollutant models for children, in addition to NO_2_ and CO within single-pollutant models for adults. Adjusted odds ratios for ED visits for asthma, according to levels of ambient air pollution for each pollutant, may be found in Appendix A. The average levels for the five year period were: PM_2.5_: 11.41 µg/m^3^, SD ± 5.95; O_3_: 38.5 ppb SD ± 15.9; SO_2_: 9.25 ppb, SD ± 10.89; NO_2_: 22.7 ppb, SD ± 9.4; and CO: 0.51 ppm, S.D. ± 0.27; and average temperature was 11.7 degrees Celsius SD ± 18.1.

### 3.2. Children Ages 5–17 Years

Statistically significant effects of O_3_ were noted for lag day 1 in the single- and two-pollutant models (OR: 1.12, 95% CI: 1.02–1.22, *p* < 0.01) in the single pollutant model; and two-pollutant model (OR: 1.10, 95% CI: 1.01–1.20, *p* < 0.01) (Figure 3). This same effect was evident in the multi-pollutant model adjusting for PM_2.5_, temperature, and the other pollutants (Appendix A). No significant positive associations were observed between SO_2_, NO_2_, and CO and ED visits. See Appendix A for complete data. Of note, PM_2.5_ in the two-pollutant model had a marginally significant protective effect, which requires further study.

### 3.3. Adults Aged 18 Years and Older

Statistically significant positive effects of CO were noted in adult ED visits for asthma on lag day 5 (OR: 1.13, 95% CI: 1.00–1.28, *p* < 0.01) and average lag days 0–5 (OR: 1.22, 95% CI: 1.00–1.49, *p* < 0.01) in the single-pollutant model. Similarly, a statistically-significant positive effect of NO_2_ was seen in adult ED visits for asthma for lag day 5 (OR: 1.04, 95% CI: 1.00–1.07, *p* < 0.01) in the single-pollutant model (Figure 4). No significant associations were observed for SO_2_ and O_3_. See Appendix A, for complete data on CO and NO_2_, respectively. In the five-pollutant model, no statistically significant positive effects were seen for any of the pollutants at any of the lags examined (Appendix A).

### 3.4. Summary

In children aged 5–17 years, there were statistically significant increases in asthma-related ED visits for O_3_ on lag day 1 (one day before the visit) in all models. Typically, this was found because of the delay in seeking care.

In adults 18 years and older, there was no statistically significant effect of ozone on asthma-related ED visits in any of the models. Instead, we found statistically significant effects of CO on lag day 5 and average lag days 0–5 and NO_2_ on lag day 5 in the single-pollutant model only.

## 4. Discussion

Our analysis showed significant associations between daily pollution levels and asthma-related ED visits for both children and adults during the five-year study period. Higher ambient levels of O_3_ were associated with increased ED visits in children, though not in adults. Increased ambient levels of NO_2_ and CO were associated with increased odds of an ED visit in adults. Additionally, our study demonstrates differences in asthma-related ED visits based upon sex. This finding is in line with a prior report that post-pubescent women have been shown to have poorer asthma outcomes compared to their male counterparts [26].

A strength of our study was the extended lagged analyses through day 5. Some previous studies examined lag days 1–3 and found no association between pollution levels and ED visits. Significant associations observed at extended lag periods in the current study (e.g., lag day 5) suggest the potential for underlying biological mechanisms as potential contributors to this delayed response. Another possibility is the potential delay in asthmatics seeking ED treatment.

It was unexpected that our study did not show an association between PM_2.5_ and ED visits in the single-pollutant model. The association of asthma exacerbations with elevations in short-term exposures to PM_2.5_ are well established [27,28,29]. Studies suggest PM_2.5_ may activate pathways associated with oxidative stress, which can increase airway hyper-responsiveness [30]. An interesting feature of PM_2.5_ is its ability to serve as a transport vessel for airborne gaseous particles to travel deep into the bronchial airway. Therefore, it is important to consider how PM_2.5_ concentrations change when combined with gaseous pollutants, as we did in our analysis.

Although the sample size used for the analysis in children was smaller than for adults, there were statistically significant increases in asthma-related ED visits for O_3_ noted on lag day 1 (one day before the visit). Typically, this lag occurs because of the delay in seeking care. This finding was noted to overcome the collinearity among pollutants after adjustment for both temperature and PM_2.5_. This finding is consistent with other studies showing an association between O_3_ and asthma-related ED visits in children. Another study showed that after adjusting for seasonal variation, high levels of O_3_ and SO_2_ were associated with asthma exacerbations in children [31]. Another study demonstrated a positive relationship between O_3_ levels and asthma-related ED visits, with associations being the strongest during the warm season [16,32,33]. In contrast to our findings in children, we did not find a positive association between O_3_ and asthma-related ED visits in adults. Consistent with our findings, an Australian study showed increased air pollution affected ED visits for children but not adults [34]. There are several explanations for these discrepant findings in children and adults. Toxicological studies have shown that children tend to breathe in more air through their mouths, as opposed to adults, who breathe primarily through their nasal passages, which help to filter the air before reaching the lungs [35]. A systematic review of 27 epidemiological studies concluded that children may be at higher risk from O_3_ because of their immature immune systems, increased durations of time spent outside, and increased air exchange relative to body mass, as compared to adults; therefore, higher exposure levels may be why we see effects from O_3_ in children when compared with adults [8].

Lag day 5 was implicated in the adult analyses of NO_2_ and CO. This lag may be related to the potential occupational exposures experienced by adults. Additional studies are needed to better understand the significance of this extended lag effect. Some studies suggest delayed effects might exist between various pollutants and asthma outcomes related to delayed physiologic responses [8]. Our finding of an association between NO_2_ exposure, which is a known marker of traffic-related pollution that has been associated with lung inflammation, and asthma-related ED visits in adults is consistent with several previous studies. NO_2_ was not implicated in asthma-related ED visits in children in our study.

Lastly, the primary effects of high outdoor CO exposures are related to hypoxia, which results in confusion, headache, and nausea [36]. CO might be a marker for other noxious combustion products, such as the burning of wood, coal, gas, and tobacco [36]. CO pollution has been associated with decreased lung function in adults with asthma [37,38,39,40]. CO was not linked with asthma ED visits in children within our study; however, a prior study reported increased odds of school-based health clinic visits related to high ambient levels of CO [41].

## 5. Study Limitations

As with all epidemiological studies, limitations exist for this study. For example, environmental factors such as elevation could not be completely controlled for and may have influenced the results. O_3_ tends to be increased at higher elevations [42]. Furthermore, we used two EPA-grade reference monitors from different locations within Allegheny County to perform the analyses. Because of this small air monitor network, we relied on regional pollution data rather than hyper-local conditions. Therefore, we were unable to capture local-scale spatial variations that may have occurred for various pollutants. The more distant monitor that provided SO_2_ concentrations may not have been representative of the entire cohorts’ ambient exposure, since the participating hospitals were clustered near the Pittsburgh reference monitor. Furthermore, due to the limited nature of our dataset, confounders such as socioeconomic status, type of health insurance, severity of asthma, and use of controller therapy were not examined. Lastly, ED visits for asthma are only one piece of a larger picture that describes asthma burden. Future studies should consider multiple endpoints (i.e., ED visits, school-based clinic visits, and outpatient physician office visits) to obtain a more comprehensive picture of how asthma control is exacerbated by air pollutants.

## 6. Conclusions

There is an association between O_3_ exposure in children and NO_2_ and CO exposure in adults and asthma-related ED visits within the greater Pittsburgh area. Public health intervention(s) aimed at mitigating the effects of air pollutants targeted to the entire population may have significant benefits for children and adults with asthma, as well as the public as a whole.

## Figures and Tables

**Figure 1 ijerph-17-08619-f001:**
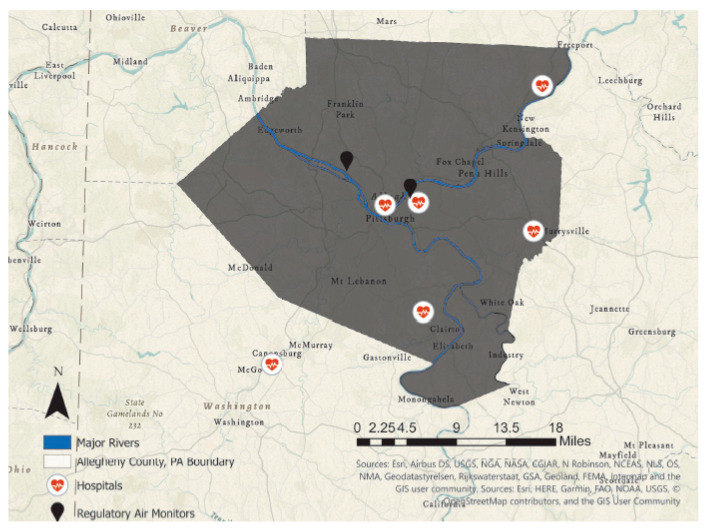
Map of Allegheny County, PA, depicting locations of regulatory air monitors and hospitals where asthma-related Emergency Department (ED) visits were recorded between 2008 and 2013.

**Figure 2 ijerph-17-08619-f002:**
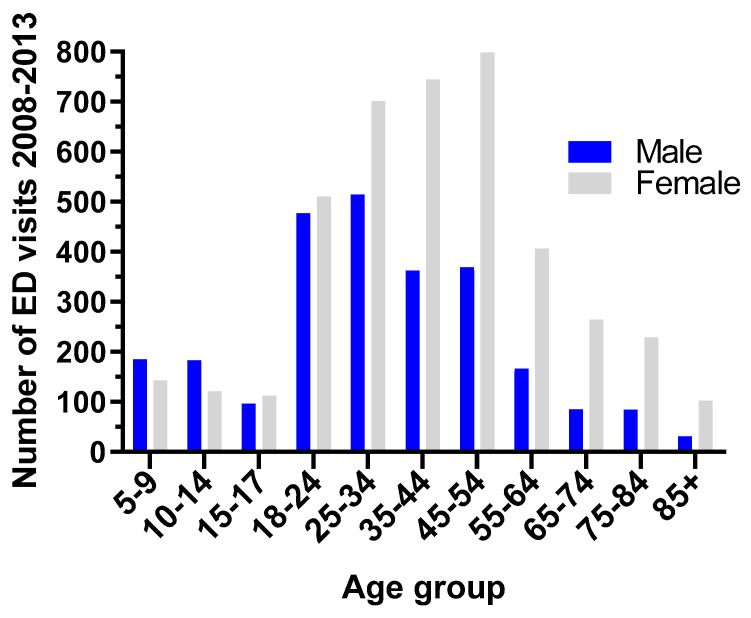
Cases of asthma-related ED visits between 2008 and 2013 separated by sex and age.

**Figure 3 ijerph-17-08619-f003:**
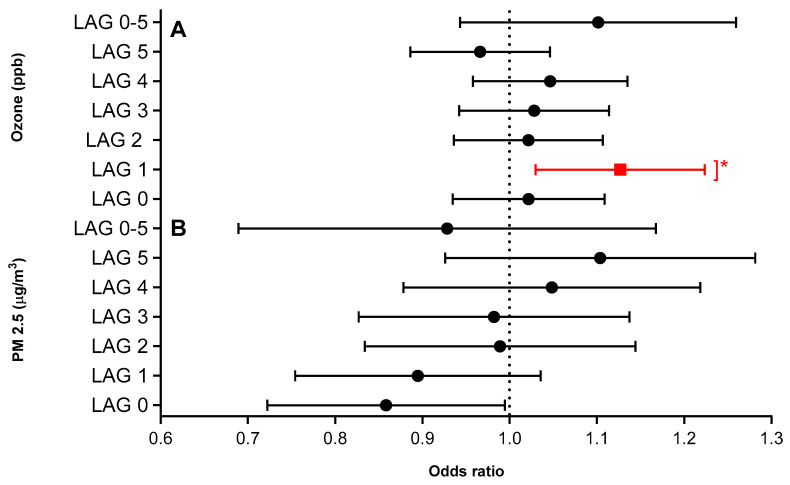
(**A**,**B**) Odds ratios of an asthma-related ED visit and significant lag days for children ages 5–17 using a two-pollutant model (single day lags and average lag, adjusted for apparent temperature (at same lag), which includes PM_2.5_ and O_3_, adjusted for temperature.) Referent time period is 28 days. * *p*-value < 0.05.

**Figure 4 ijerph-17-08619-f004:**
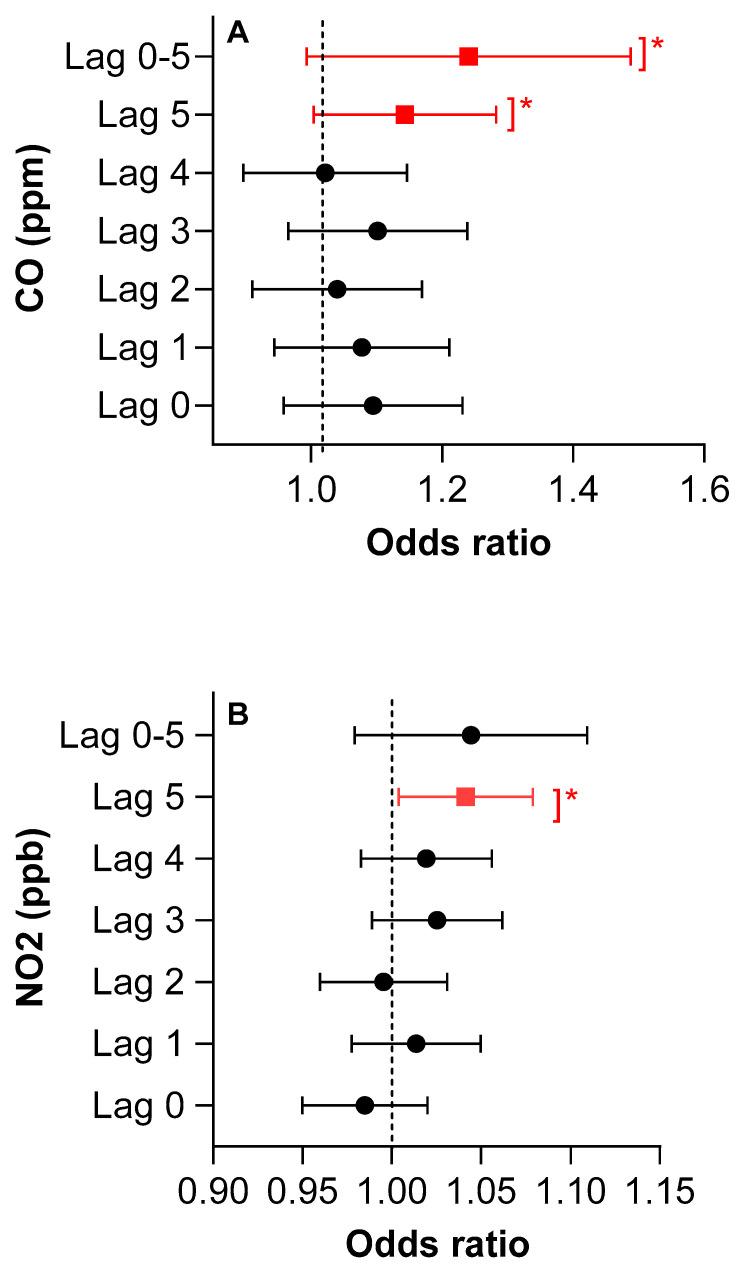
(**A**,**B**) Odds ratios of an asthma-related ED visit and significant lag days for adults aged 18 and older using a single-pollutant model for CO and NO_2_, adjusted for temperature. Referent time-period is 28 days. * *p*-value < 0.05.

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
