# Peer review of "Lagged Association of Ambient Outdoor Air Pollutants with Asthma-Related Emergency Department Visits within the Pittsburgh Region"

_ijerph, 2020, doi:10.3390/ijerph17228619_

Round 1
Reviewer 1 Report
This study used case-crossover analysis to understand associations between criteria air pollutants and asthma ED visits in Pittsburgh area. This study reported different associations with different pollutants between children and adult models. This difference was described for the pollutants that had association with increased ED visit, but those that showed protective effects were not mentioned. More comprehensive explanation on results is necessary. The discussion section could have better interpretation of unique study results in the context of previous studies.
Abstract:
Very general conclusion. Specific finding from the study should be emphasized. Spatial resolution mentioned only at the end and not tied to the results. Study description should include the geolocation and timeframe of this study.
Introduction:
The introduction needs to cite other previous studies examined lag effects and describe how this study provides new information to improve the understanding.
Methods:
Line 62: The case-crossover design is commonly chosen to investigate transient effects on the risk of acute health events. --> this statement needs citations.
Line 93: why calculate the average daily temperature from min and max but not using entire data points?
Results:
Line 126-8: figure 1 and 2 are switched.
Figure 3. caption says *= P-value >0.05. this should be “P-value < 0.05”. Also, what is the referent time for this result? Did it have any influence in modeling result? If so, what would be the reason? Same comment for the Figure 4.
Line 146: No association for SO2, NO2, and CO. –-> it would be nice to briefly mention that these are for single model or multiple model or both as well as the result of PM2.5 in single model.
Figure 3 also shows protective effect of PM2.5 in the two-pollutant model for Lag 0, which is very interesting and should be explained. It would be great if the authors could describe any specifics about air pollution data distribution between O3 and PM2.5. In the supplemental data, CO was showing borderline protective effect for children for Lag 2 in the multi-pollutnat model.
More importantly, for adult, ozone showed protective effects consistently in the single model for lag 0 through 4 and lag 0-5, two-pollutant models with PM2.5 and some of multipollutant models, which is opposite to the children model and contrary to current understanding of adverse respiratory effects of ozone. However, this result and interpretation were not found in the manuscript. Please explain what would be the reason we see the opposite direction of association between children and adult for ozone and describe how these findings are similar/different as compared to previous studies results.
Figure 4. what about the results from multi-pollutant models including PM2.5? it would be helpful if the multi-pollutant model results were mentioned in the main manuscript.
Females older than 25 years had higher ED visits. Any associations with air pollutants that are different from males? There are no result reported on sex, which would have been an interesting and important aspect.
What are the results from different referent times? Did different referent time result in different modeling outcomes? The supplemental material shows only 28-day referent time model.
Line 190: The authors mentioned that the lag occurs because of the delay in seeking care. How the cohort-specific culture/practice in seeking medical care for those emergency situation would play a role in different lag times reported by various studies? Would type of insurance and SES and ED rate of this cohort be different from other areas where shorter lag days are reported?
Author Response
Thank you for your comments/feedback. Please see our responses below.
General Comment: This study used case-crossover analysis to understand associations between criteria air pollutants and asthma ED visits in Pittsburgh area. This study reported different associations with different pollutants between children and adult models. This difference was described for the pollutants that had association with increased ED visit, but those that showed protective effects were not mentioned. More comprehensive explanation on results is necessary. The discussion section could have better interpretation of unique study results in the context of previous studies.
Response: Added the following statement on lines 157-158. “Note some pollutants may have a protective effect, however this is poorly studied and further study is needed.”
Additional explanation of results has been added to line 183-198. Conclusion (Children 5-17): Although the sample size used for the analysis in children was not large, there were statistically significant increases in asthma exacerbations (ED visits) for ozone noted on lag day one, (one day before the visit). Typically this is found because of the delay in seeking care. This finding was noted after adjustment for both temperature and PM2.5 in two pollutant models as well as for temperature and the other four pollutants. The implication of these results is that even though there have been decreases in the levels pollution in the greater Pittsburgh area, there still remains an association between ozone exposure and asthma exacerbations ( ED visits) in the greater Pittsburgh area.
Conclusion (Adults 18+): Unlike the significant effects of ozone noted on lag day 1 for asthma ED visits in children, we noted no significant increase in asthma ED visits among adults in our study population for ozone. Adjusting for temperature, statistically significant positive effects of carbon monoxide were noted on adult ED visits for asthma for lag day 4 ( OR-1.137 , 95%CI, 1.007-1.283) and average lag 0-5 ( OR = 1.224, 95% CI, 1.002-1.496) Additionally, as noted in the results for adults 18+ for NO2 (single pollutant models), adjusted for temperature); statistically significant positive effects of NO2 were seen on adult ED visits for asthma for lag day 5, (OR=1.04, 95% CI, 1.004-1.074).
Abstract: Very general conclusion. Specific finding from the study should be emphasized. Spatial resolution mentioned only at the end and not tied to the results. Study description should include the geolocation and timeframe of this study.
Response: Modified conclusion to Line 28-30 “Despite many fossil fuel air pollutant levels waning over time, ozone has not declined in the greater Pittsburgh region. This indicates additional mitigation efforts are needed to protect vulnerable children with asthma.”
Introduction: The introduction needs to cite other previous studies examined lag effects and describe how this study provides new information to improve the understanding.
Response: Added the following citations
Fauroux, B., Sampil, M., Quénel, P. & Lemoullec, Y. Ozone: A trigger for hospital pediatric asthma emergency room visits. Pediatr. Pulmonol. 30, 41–46 (2000).
Gleason, J. A. & Fagliano, J. A. Associations of daily pediatric asthma emergency department visits with air pollution in Newark, NJ: Utilizing time-series and case-crossover study designs. J. Asthma 52, 815–822 (2015).
And the following text added to lines 58-63 “Our findings implicate O3 in both single and multipollutant models. Other studies have also found similar associations between ED visits and O3 [15,16] . However, these studies do not include five pollutant models. A strength of our study is the pollutants in our adjusted model as very few studies examine five-pollutant models in conjunction with asthma ED visits. The primary objective of this study was to examine the risk of asthma related ED visits in both children and adult populations relative to ambient air pollution.”
Methods: Line 62: The case-crossover design is commonly chosen to investigate transient effects on the risk of acute health events. --> this statement needs citations.
Response: Added the following citation- Jaakkola, J. J. K. Case-crossover design in air pollution epidemiology. in European Respiratory Journal, Supplement 21, 81–85 (European Respiratory Society, 2003).
Line 93: why calculate the average daily temperature from min and max but not using entire data points?
Results:
Line 126-8: figure 1 and 2 are switched.
Response: Omitted Figure 2 reference and corrected Figure 1.
Figure 3. caption says *= P-value >0.05. this should be “P-value < 0.05”. Also, what is the referent time for this result? Did it have any influence in modeling result? If so, what would be the reason? Same comment for the Figure 4.
Response: Referent time period is 28 days. Corrected >0.05 to <0.05 in both figures. Clarified figure legend to Line 161-162 “using a two-pollutant model (single day lags and average lag, adjusted for apparent temperature (at same lag), which includes PM2.5 and O3, adjusted for temperature.”
Line 146: No association for SO2, NO2, and CO. –-> it would be nice to briefly mention that these are for single model or multiple model or both as well as the result of PM2.5 in single model.
Response: Added to lines 150-152 “The strongest associations between pollutant levels and ED visits were observed for O3 within single, double, and five pollutant models, both with and without PM2.5, NO2 and CO both within single pollutant models.”
Figure 3 also shows protective effect of PM2.5 in the two-pollutant model for Lag 0, which is very interesting and should be explained. It would be great if the authors could describe any specifics about air pollution data distribution between O3 and PM2.5. In the supplemental data, CO was showing borderline protective effect for children for Lag 2 in the multi-pollutnat model.
Response: Running the number of lags and pollutants for air pollution models of this kind, it is not unusual to show a protective effect for some of the outcomes, which is often unexplainable and spurious. Similarly, we see one and two-day lag effects which are explained by delay in care seeking in any 24-hour period prior to an exacerbation. As ozone is markedly lower in the winter, there are typically statistically significant effects during the warm months but not the cold periods. I cannot give any specific explanation for the protective effect of Pm2.5 for lag 0.
More importantly, for adult, ozone showed protective effects consistently in the single model for lag 0 through 4 and lag 0-5, two-pollutant models with PM2.5 and some of multipollutant models, which is opposite to the children model and contrary to current understanding of adverse respiratory effects of ozone. However, this result and interpretation were not found in the manuscript. Please explain what would be the reason we see the opposite direction of association between children and adult for ozone and describe how these findings are similar/different as compared to previous studies results.
Response: Added the following to lines 252-255 A systematic review of 27 epidemiological studies concluded that children may be a higher risk from O3 because of their immature immune systems, increased durations of time spent outside, increased air exchange relative to their body mass compared with adults, and therefore higher exposure levels may be why we see effects from O3 in children compared with adults [29]. Added citation Rodriguez-Villamizar, L. A., Magico, A., Osornio-Vargas, A. & Rowe, B. H. The effects of outdoor air pollution on the respiratory health of Canadian children: A systematic review of epidemiological studies. Canadian Respiratory Journal 22, 282–292 (2015).
Response: Added the following to Line 174-176. Five pollutant model (PM2.5, Ozone, CO, NO2, SO2): Table 15 presents the results for adults for the five pollutant models, adjusted for temperature. With all five pollutants in the models, no statistically significant positive effects were seen for any of the pollutants at any of the lags examined.
Females older than 25 years had higher ED visits. Any associations with air pollutants that are different from males? There are no result reported on sex, which would have been an interesting and important aspect.
Response: Unfortunately, we did not consider gender as an effect modifier but I agree it would have been an interesting aspect to consider.
What are the results from different referent times? Did different referent time result in different modeling outcomes? The supplemental material shows only 28-day referent time model.
Response: the referent times are 7,14 or 21 days before or after the incident emergency room visit, this allows the case to be her or his own control and takes care of seasonal and day of week bias, see https://www.ncbi.nlm.nih.gov/pmc/articles/PMC2805787/ This design uses only cases and compares each individual’s exposure experience just prior to the event with exposure at other time periods, the referent periods. For our analyses, the level of ambient air pollution in the time period just before an individual visited the Emergency Department was compared with levels at referent time periods. This is one of the primary strengths of the case crossover method as it adjusts for not only day of the week but also to an extent season and location. The standard referent model is based on the 28 referent model as detailed in the article by Haley, Talbot and Talbott.
Line 190: The authors mentioned that the lag occurs because of the delay in seeking care. How the cohort-specific culture/practice in seeking medical care for those emergency situation would play a role in different lag times reported by various studies? Would type of insurance and SES and ED rate of this cohort be different from other areas where shorter lag days are reported?
Response: We agree that care seeking behavior and ability to pay for care are crucial questions, unfortunately the study did not have access to the type of data needed to answer these questions, we did not get access to payor status and the SES of the family was not available , only the age, gender and zipcode of residence so it was not possible to access adequate information to answer these questions. A look at other countries dependent on the type of medical care access and air pollution studies of asthma might be worth a look. You could argue that having proper PCP medical access on a routine basis would cut down immeasurably the use of ED departments as well.
Reviewer 2 Report
Suggest a paragraph description on sources of pollutants, dispersion mechanism of the pollutants from the source to the regions of space, and a comparison (or cause and effect scenario) of epa standards attainment with increase asthma burden.
Author Response
Thank you for your feedback. Please see our comment below
Suggest a paragraph description on sources of pollutants, dispersion mechanism of the pollutants from the source to the regions of space, and a comparison (or cause and effect scenario) of epa standards attainment with increase asthma burden.
Response: This is a great suggestion, thank you. Unfortunately due to the small EPA monitoring network used in our analyses relative to the size of the county, it would be too difficult to refer to specific sources that may or may not contribute. Additionally, specific data on factory emissions would be needed to address this thoroughly.
Reviewer 3 Report
Abstract: The description of the results should be improved (please use expressions like: “...was significantly associated with...” or “a significant association was found for...”).
Lines 36: reference 1 is indicated as a superscript.
Lines 45-46: are such inconsistencies general or specific for asthma admission? Is the paper intended to address such inconsistencies? In general, the added value of the manuscript does not clearly emerge from the introduction.
Line 82: please delete the point after SO2 and rewrite “Measurement” lowercase.
Lines 102-103: “With this approach, the overall time period is divided into strata and exposure in the time just prior to the event”: this sentence appears unclear or incomplete.
Lines 106-107: the “washout” period was 7 days. What if two events on the same individuals were 14 or 21 days apart? Did the days of admission overlap with referent (control) days in the same stratum in that cases?
Line 112: unconstrained lag models are known to be prone to serious multicollinearity concerns. Did the Authors worry about that?
There is no mention about multi-pollutant models in the “statistical analysis” section.
How was temperature effect modeled? A linear relationship is not appropriate; a non linear relationship (e.g. a cubic spline) would be more appropriate to account for both cold and heat effects.
Lines 121-124: please move to the “statistical analysis” section.
Lines 137: please check this sentence.
Supplementary tables: are all the lags included in the same models (unconstrained distributed lag models)? From lines 111-113it appears that models were first estimated including just one single day lag at a time and then including all the lags together, but there appears to be no distinction about these two model settings in the results. Moreover, what is “AvgTemp_F_INT” in Table S8?
Author Response
Thank you for your feedback and thoughtful comments. Please see our responses below.
Abstract: The description of the results should be improved (please use expressions like: “...was significantly associated with...” or “a significant association was found for...”).
Response: In trying to stay within the strict 200 word abstract limit, redundant verbiage was omitted.
Lines 36: reference 1 is indicated as a superscript.
Response: removed superscript
Lines 45-46: are such inconsistencies general or specific for asthma admission? Is the paper intended to address such inconsistencies? In general, the added value of the manuscript does not clearly emerge from the introduction.
Response: This paper aims to focus specifically to ED visits and pollution. Inconsistencies related to air pollution and asthma specifically. This may be due to inconsistent methods, population and locations studied.
Line 82: please delete the point after SO2 and rewrite “Measurement” lowercase.
Response: Replaced “Measurement” with measurement
Lines 102-103: “With this approach, the overall time period is divided into strata and exposure in the time just prior to the event”: this sentence appears unclear or incomplete.
Response: You are correct! It was cut off and should read: “With this approach, the overall time period is divided into strata and exposure in the time period just prior to the event and exposures in multiple reference periods is compared within that stratum of time.” This has been inserted in its place on lines, 102-104.
Lines 106-107: the “washout” period was 7 days. What if two events on the same individuals were 14 or 21 days apart? Did the days of admission overlap with referent (control) days in the same stratum in that cases?
Response: Theoretically they would be considered a “new event” Each referent control day is unique to each case but are not mutually exclusive to other cases.
Line 112: unconstrained lag models are known to be prone to serious multicollinearity concerns. Did the Authors worry about that?
Response: We have removed the line about unconstrained lag models as we did not include this in the original report and do not have results for this set of models.
There is no mention about multi-pollutant models in the “statistical analysis” section.
Response: We controlled for temperature effects in all models by including average temperature at the same lag(s) as the pollutant(s). In addition to single pollutant models for PM2.5, ozone, NO2, SO2, and CO, we also conducted two-pollutant models for PM2.5 and ozone adjusted for temperature. Lastly, we conducted multi-pollutant models with all five pollutants.
How was temperature effect modeled? A linear relationship is not appropriate; a non linear relationship (e.g. a cubic spline) would be more appropriate to account for both cold and heat effects.
Response: Studies that used time-stratified CCO typically select a control day on the same day of the week during the same month as the event, although other schemes (e.g., selecting days during the same month with comparable temperature) are also used. Studies that used unidirectional CCO designs used a variety of schemes to select control days (e.g., day 7 before the event). Because case crossover is so tightly controlled by virtue of the referent definition of within 28 days, the use of cubic splines was not used. Temperature was included in each model as an adjustment variable however.
Lines 121-124: please move to the “statistical analysis” section.
Response: We have removed from results and inserted into analysis section. Thank you
Lines 137: please check this sentence.
Response: This sentence was removed
Supplementary tables: are all the lags included in the same models (unconstrained distributed lag models)? From lines 111-113it appears that models were first estimated including just one single day lag at a time and then including all the lags together, but there appears to be no distinction about these two model settings in the results. Moreover, what is “AvgTemp_F_INT” in Table S8?
Response: As stated previously we did not run unconstrained distributed lags as shown in the tables, there were three sets of analyses: one set Single Pollutant, ALL FIVE POLLUTANTS, Single day lags and average lag, Adjusted for Apparent Temperature , another set for children and adults: Two Pollutants,( PM2.5 and ozone) , Single day lags and average lag, Adjusted for Apparent Temperature (at same lag) analyses: 7-day washout; 28-day referents and a third set: for PM2.5 (per 10 µg/m3), Ozone (per 10 ppb), CO (ppm), NO2 (per 10 ppb), SO2 (per 10 ppb). Multiple Pollutant models, Single day lags and average lag, Adjusted for Apparent Temperature (at same lag) analyses: 7-day washout; 28-day referents
AvgTemp_F_INT” in Table S8 has been corrected to AvgTemp. Thank you!
Reviewer 4 Report
Review of the Research Article 969220 titled " Lagged Association of Ambient Outdoor Air Pollutants on Asthma Related Emergency Department Visits within the Pittsburgh Region ".
The submitted manuscript describes a study investigating the short-term effects of ambient outdoor air pollution on asthma related emergency visits. This manuscript could potentially make a contribution to the literature but needs more detail in the introduction, methods, results and discussion to be accepted for publication. More specific comments are made below.
Abstract
- The research gap which the study investigated should be stated.
Introduction
- The authors should clearly describe the research gap the study was addressing and new evidence the study intended to produce.
Materials and methods
- More detail should be provided about the study area (county, population size, sources of air pollution) and health care facilities (size, capacity) where cases were extracted. Figure 1 should be cited in this section.
- In section 2.1, more detail should be provided on the air pollution monitors measurement specifications.
- The statistical analysis section (section 2.2) only describes the approach in handling the lag period. The univariate and bivariate analysis should be mentioned and the detail of multivariate analysis described. Which co-variates were considered and why was only temperature included in statistical models? Which two pollutant models and multipollutant models were tested, in addition to the single pollutant models.
- The incorrect figures are cited in lines 126 and 127.
- The descriptive statistics of the air pollution levels and climate variables should be presented.
- The meaning of the sentence in lines 138 -140 “The overall effect of an effect of exposure …” is unclear.
- The basis for including some results in the main paper and others in the supplementary results should be clarified. Why was the results of the 2 pollutant model presented in figure 3 and a single pollutant in figure 4? Was the findings of the multipollutant models?
- A summary statement should be included stating what results are presented in the supplementary results including which single pollutants and multi-pollutant results and the confounders included.
Discussion
- Th findings appears overstated. The authors should explain why results are not consistent in 2 pollutant and multipollutant models. As mentioned before, the inclusion of covariates is not discussed. The authors should investigate if the significant results could be due to chance ie because of multiple comparisons.
- More detail should be provided about other comparative studies in the literature ie sample size, air pollution levels and if single and 2 pollutant models were applied.
- In the limitations, confounders not measured in the study should be mentioned.
- Apart from the location of air monitor, the limitations in using air pollution data from stationary air monitors should be mentioned in relation to other more sophisticated methods.
Conclusions
- As indicated before, the findings are overstated and should be toned down and the new evidence should be highlighted.
Author Response
Thank you for your time, effort, and feedback. Please see our responses below.
Abstract
The research gap which the study investigated should be stated.
Response: Despite many fossil fuel air pollutant levels waning over time, ozone has not declined in the greater Pittsburgh region. This indicates additional mitigation efforts are needed to protect vulnerable children with asthma.
Introduction
The authors should clearly describe the research gap the study was addressing and new evidence the study intended to produce.
Response: Line 58-59 The primary objective of this study was to examine the risk of asthma related ED visits in both children and adult populations relative to ambient air pollution. Very few studies examine multi-pollutant models that include five pollutants.
Materials and methods
More detail should be provided about the study area (county, population size, sources of air pollution) and health care facilities (size, capacity) where cases were extracted. Figure 1 should be cited in this section.
Response: Added the following to Lines 72-76. Allegheny Health Network were assembled. This hospital system included facilities that were part of a six hospital regional health system. All have hospital-based emergency departments staffed by emergency physicians with up-to-date diagnostic and treatment capabilities. The annual visits to these EDs ranged from 25,000 to 55,000 per year.
In section 2.1, more detail should be provided on the air pollution monitors measurement specifications.
Response: Added the following to lines 88-89 These monitoring locations are part of the EPA Air Quality System (AQS) that is used to monitor compliance with the Clean Air Act.
The statistical analysis section (section 2.2) only describes the approach in handling the lag period. The univariate and bivariate analysis should be mentioned and the detail of multivariate analysis described. Which co-variates were considered and why was only temperature included in statistical models? Which two pollutant models and multipollutant models were tested, in addition to the single pollutant models.
Response: The covariates of age, gender, race and SES and other time invariant covariates are all matched as the case crossover design by its inherent nature uses the case as its own control. Using referent periods at7, 14 or 21 days earlier or later controls for day of week effects and choosing referents in a 28 day period helps controls for seasonal effects. This is the strength of the case cross over analysis for an acute event.
The incorrect figures are cited in lines 126 and 127.
Response: Figures cited have been corrected. Thank you!
The descriptive statistics of the air pollution levels and climate variables should be presented.
Response: Added to lines 151-153 “The average levels for the five year period were: PM2.5: 11.41 µg/m3, SD ± 5.95, 53.5, S.D. ±18.14, O3: 38.5ppb ±15.9, SO2: 9.25 ppb, S.D. ±10.89 , NO2: 22.7 ppb, S.D.: ±9.4, CO: .51 ppm, S.D.: ±.27 and average temperature was 53.1 degrees Fahrenheit, SD± 18.1.”
The meaning of the sentence in lines 138 -140 “The overall effect of an effect of exposure …” is unclear.
Response: lines 138 -140 The overall effect of an increase in exposure per increment of 10 was changed to “The odds of an asthma ED visit related to the effect of an increase in exposure”
The basis for including some results in the main paper and others in the supplementary results should be clarified. Why was the results of the 2 pollutant model presented in figure 3 and a single pollutant in figure 4? Was the findings of the multipollutant models?
Response: Significant findings were included in the main text. Due to the large amount of data, non significant results were kept to the supplementary section. Included the following on lines 183-186 “Five pollutant model (PM2.5, Ozone, CO, NO2, SO2): Table 14 presents the results for adults for the five pollutant models, adjusted for temperature. With all five pollutants in the models, no statistically significant positive effects were seen for any of the pollutants at any of the lags examined.”
For the children analyses the following references the five pollutant model Lines 157-160.
“the Statistically significant effects of O3 were noted for Lag day 1 (Figure 3) in the two pollutant model, the ozone remained significantly related to ozone exacerbations and ER visits on lag day 1, (p<.01,OR of 1.12 ( 95% CI, 1.026- 1.22). This was also evident in the multi pollutant models adjusting for PM2.5, temperature and the other four pollutants (See Supplementary section S1 Table S14).”
A summary statement should be included stating what results are presented in the supplementary results including which single pollutants and multi-pollutant results and the confounders included.
Response: The following statement was added to lines 127-128. Only significant findings are included within the text, however the full data set may be found within the supplementary tables S1-S14.
Discussion
The findings appears overstated. The authors should explain why results are not consistent in 2 pollutant and multipollutant models. As mentioned before, the inclusion of covariates is not discussed. The authors should investigate if the significant results could be due to chance ie because of multiple comparisons.
Response: Thank you for the important question: The covariates of age, gender, race and SES and other time invariant covariates are all matched as the case crossover design by its inherent nature uses the case as its own control. You can consider effect modification through a stratified analysis but essentially the covariates above are taken care of in this design. Using referent periods at 7, 14 or 21 days earlier or later controls for day of week effects and choosing referents in a 28 day period helps controls for seasonal effects. This is the strength of the case cross over analysis for an acute event. Temperature was a primary adjustment however, The data for children asthma ED visits and ozone in this set of analyses is very consistent: exposure on lag day one for children, the results of the single pollutant model (,1.105 95% CI 1.017 1.200) ,double pollutant model (OR:1.124 95%CI 1.031 1.225) ( with Pm2.5 adjusting for temperature) and multipollutant models, OR 1.120 95% 1.026 1.222) The odds ratios were consistently similar at lag day 1 , regardless of model or adjustment which is very significant. There was a five to 12% increase in ED visits on lag day one for each 10 ppb of ozone for children, this was not seen for adults however, it is well known that children are more vulnerable to allergens and pollutants and spend more time out of doors.
Like wise the relationship of CO and asthma exacerbations in adults shows similar consistency whether a single, or multipollutant model at the same lag. There were no significant effects of ozone for the adults.
These were primary findings and we believe are important. Multiple comparisons are always an issue ; however, that is one reason why the we look across the various single, double and multiple models to look for consistency across time ( lags) and the two primary age groups.
More detail should be provided about other comparative studies in the literature ie sample size, air pollution levels and if single and 2 pollutant models were applied.
Response: Very few studies actually look at the five pollutant model. This was a strength of our study. the following text added to lines 58-63 “Our findings implicate O3 in both single and multipollutant models. Other studies have also found similar associations between ED visits and O3 [15,16] . However, these studies do not include five pollutant models. A strength of our study is the pollutants in our adjusted model as very few studies examine five-pollutant models in conjunction with asthma ED visits. The primary objective of this study was to examine the risk of asthma related ED visits in both children and adult populations relative to ambient air pollution.”
In the limitations, confounders not measured in the study should be mentioned.
Response: Line 278-280 the following sentence was added. “Furthermore, because of our limited dataset, confounders such as socioeconomic status, type of health insurance, or severity of asthma may have influenced our results.”
Apart from the location of air monitor, the limitations in using air pollution data from stationary air monitors should be mentioned in relation to other more sophisticated methods.
Response: Our analysis assumes that the downtown monitor is an indicator of the temporal trends in regional pollution. E.g., if that monitor is high today, then all monitors will be high(er) than normal that day. I think this is more-or-less correct. PM2.5 is a regional pollutant (lots of papers on this). O3 is generally spatially homogeneous. The downside is that you don’t capture local-scale spatial variations, or if there is a hotspot in one location on a given day. Not sure what is intended by “more sophisticated” methods. Something like a LUR gives you a spatial pattern, but using this we lose the temporal aspect that our study relies on.
Response: Added the following to lines 275-276 “Because of this small air monitor network, we relied on regional pollution data rather than hyper-local conditions. Therefore we are unable to capture local-scale spatial variations that may occur for various pollutants.”
Conclusions
As indicated before, the findings are overstated and should be toned down and the new evidence should be highlighted.
Response: Additional explanation related to O3 and children along with references has been included on lines 252-255 “A systematic review of 27 epidemiological studies concluded that children may be a higher risk from O3 because of their immature immune systems, increased durations of time spent outside, increased air exchange relative to their body mass compared with adults, and therefore higher exposure levels may be why we see effects from O3 in children compared with adults” additional sources listed
Rodriguez-Villamizar, L. A., Magico, A., Osornio-Vargas, A., & Rowe, B. H. (2015). The effects of outdoor air pollution on the respiratory health of Canadian children: A systematic review of epidemiological studies. Canadian respiratory journal, 22(5), 282–292. https://doi.org/10.1155/2015/263427
Round 2
Reviewer 1 Report
The authors improved the manuscript. There are few minor things to be corrected/clarified.
Line 28-40: the authors added these sentences “ Despite many fossil fuel air pollutant levels waning over time, ozone has not declined in the greater Pittsburgh region. This indicates additional mitigation efforts are needed to protect vulnerable children with asthma…”
Does ozone level still the same as before? Or are authors trying to say the association between ozone and ED visit still remains significant despite the ozone level reduction? This should be clarified.
Figure 1 and Figure 2 call-outs need to be corrected.
Figure 3: please indicate in the figure caption that Referent time period is 28 days.
Few of the newly added sentences with track change are not complete sentences.
Author Response
Reviewer 1 Review Report (RD 2)
Lines 28-30 Line 28-40: the authors added these sentences “ Despite many fossil fuel air pollutant levels waning over time, ozone has not declined in the greater Pittsburgh region. This indicates additional mitigation efforts are needed to protect vulnerable children with asthma…”
Does ozone level still the same as before? Or are authors trying to say the association between ozone and ED visit still remains significant despite the ozone level reduction? This should be clarified.
Response: Ozone levels have worsened over time. Lines 28-30 changed to the following “Despite many fossil fuel air pollutant levels waning over time, ozone has increased in the greater Pittsburgh region. This indicates additional mitigation efforts are needed to protect vulnerable children with asthma.
Figure 1 and Figure 2 call-outs need to be corrected.
Response: Thank you!
Line 96 “Figure 1” changed to “Figure 2”
Line 85 omitted “Figure 2”
Figure 3: please indicate in the figure caption that Referent time period is 28 days.
Response: Lines 172-173 added “Referent time-period is 28 days.” Also added “Referent time-period is 28 days.” To Figure 4. Lines 187-188.
Few of the newly added sentences with track change are not complete sentences.
Response:
Lines 58-61 changed “Our findings implicate O3 in both single and multipollutant models. Other studies have also found similar associations between ED visits and O3 [18,19] . However, these studies do not include five pollutant models. A strength of our study is the pollutants in our adjusted model as very few studies examine five-pollutant models in conjunction with asthma ED visits.”
to
“Our findings implicate O3 in both single and multipollutant models. Other studies have found similar associations between ED visits and O3 [18,19], however, these studies do not include five pollutant models. Few studies examine five-pollutant models in conjunction with asthma ED visits.”
Lines 75-79 changed from “Data on ED visits, with a primary discharge diagnosis of respiratory disease from July 1, 2008 to June 30, 2013, from Allegheny Health Network included facilities that were part of a six hospital regional health system. All have hospital-based emergency departments staffed by emergency physicians with up-to-date diagnostic and treatment capabilities. The annual visits to these EDs ranged from 25,000 to 55,000 per year.”
to
“Data on ED visits, with a primary discharge diagnosis of respiratory disease from July 1, 2008 to June 30, 2013, from Allegheny Health Network (AHN) were included in our analysis. AHN included facilities that were part of a six hospital regional health system. All have hospital-based emergency departments staffed by emergency physicians with up-to-date diagnostic and treatment capabilities. The annual visits to these EDs ranged from 25,000 to 55,000 per year.
Lines 152-154 changed from “The strongest associations between pollutant levels and ED visits were observed for O3 within single, double, and five pollutant models both with and without PM2.5, and NO2 and CO both within single pollutant models.”
to
“The strongest associations between pollutant levels and ED visits were observed for O3 within single, double, and five pollutant models. Models were devised with and without PM2.5, for NO2, and CO.
Reviewer 3 Report
Actually, some of my previous concerns were not fully addressed, in particular:
- In general, the added value of the manuscript does not clearly emerge from the introduction.
- How was temperature effect modeled? A linear relationship is not appropriate; a non linear relationship (e.g. a cubic spline) would be more appropriate to account for both cold and heat effects.
However, the Authors provided their justification, and the overall manuscript quality should not be much affected.
Author Response
Reviewer Comment: In general, the added value of the manuscript does not clearly
emerge from the introduction.
Response: Lines 58-62 have been revised to address this concern and now state “This prior study used a two-pollutant model and did not assess differential effects based on age. Other studies have found similar associations between ED visits and O3 [17,18,19]; however, few of these studies included five pollutant models. The primary objective of this study was to examine the association between ambient air pollution and the risk of asthma related ED visits in both children and adults using a five pollutant model.”
Reviewer Comment: How was temperature effect modeled? A linear relationship is not appropriate; a non linear relationship (e.g. a cubic spline) would be
more appropriate to account for both cold and heat effects.
Response: Lines 122-124 now contain a sentence stating a cubic spline was not used. This same sentence also discusses the rationale for not using a cubic spline. “Because case crossover is so tightly controlled by virtue of the referent definition of within 28 days, the use of cubic spines was not used.”